The line follower robot: a meta-analytic approach

http://orcid.org/0000-0003-4415-7416 Brigido Williamson Johnny Hatzinakis williamsonbrigido@ita.br
http://orcid.org/0000-0002-7803-1718 de Oliveira Jose M. Parente
Department of Computer Science, Aeronautics Institute of Technology , São Jose dos Campos, São Paulo , Brazil
Coelho Paulo Jorge
Electronic publication date: 2025 Mar 19
Publication date: 2025
Volume: 11
Electronic Location ID: e2744
Received 2024 Aug 23; Accepted 2025 Feb 11
Copyright: © 2025 Brigido and de Oliveira
Copyright year: 2025
Copyright holder: Brigido and de Oliveira
License: This is an open access article distributed under the terms of the Creative Commons Attribution License, which permits unrestricted use, distribution, reproduction and adaptation in any medium and for any purpose provided that it is properly attributed. For attribution, the original author(s), title, publication source (PeerJ Computer Science) and either DOI or URL of the article must be cited.
License URL: https://creativecommons.org/licenses/by/4.0/

Keywords: Line follower robot, TEMAC, Meta-analytic

Funding: The authors received no funding for this work.

==============================
Line-follower robots represent a critical segment in autonomous robotics, with broad applications ranging from industrial automation to educational tools. This meta-analytic review synthesizes research on line-follower robots, addressing a noticeable gap in the literature where comprehensive analyses are scarce. The review leverages the Theory of the Consolidated Meta-analytic Approach (TEMAC) to systematically explore 287 documents spanning from 2001 to 2024, highlighting key contributions, trends, and gaps in the field. Through this analysis, it becomes evident that while significant advancements have been made in control strategies, sensor integration, and noise reduction techniques, the literature still lacks comprehensive studies on the scalability of these technologies, especially in large-scale industrial environments. Recent research trends emphasize integrating artificial intelligence and machine learning into line-follower robots, indicating a shift towards more sophisticated, adaptable systems. Despite these advancements, challenges remain in addressing environmental variability, improving real-time adaptability, and exploring novel applications in dynamic environments. This review not only maps the historical evolution and current state of line-follower robots but also identifies future research directions that could drive the next generation of robotic systems. The findings offer valuable insights for researchers, engineers, and educators aiming to enhance the efficiency, reliability, and application scope of line-follower robots.

Introduction

Line-following robots represent a significant area of study within the broader field of autonomous robotics, with applications ranging from industrial automation to educational tools and competitive robotics. These robots, designed to follow a predefined path autonomously, are crucial for tasks that require precise navigation along designated routes, such as in automated warehouses, assembly lines, and delivery systems. The versatility and simplicity of line-following robots make them an essential component in developing more complex robotic systems, and they serve as a foundation for advancements in mobile robotics, sensor integration, and control algorithms.

The authors carried out this systematic review when they developed a simulator to test the application of an algorithm inspired by the article on simulation of line-following robots (Brigido, Maximo & Parente de Oliveira, 2024).

Despite the widespread use and importance of line-following robots, there is a noticeable gap in the literature regarding comprehensive reviews that synthesize the existing body of research. Only one systematic review, “Robotic Mobile Fulfillment System: A Systematic Review” (Benavides-Robles et al., 2024), has been identified. This review focused specifically on robotic mobile fulfillment systems (RMFS) within warehouse settings, where line-following robots are pivotal in transporting goods. However, the review did not extend its scope to encompass the diverse applications and technological advancements in line-following robots beyond the RMFS context.

This meta-analytic literature review seeks to fill this gap by comprehensively synthesizing research on line-following robots across various domains. Unlike the review mentioned above, which was limited to warehouse automation, this review covers a broader range of studies, including those focused on sensor integration, control strategies, noise reduction techniques, and applications in different environments. By analyzing the contributions, methodologies, and outcomes of multiple studies, this review aims to offer a holistic understanding of the current state of line-following robot technology, identify trends and challenges, and propose directions for future research.

The studies included in this review range from applying advanced control algorithms such as sliding mode control (Yildiz et al., 2020) to integrating computer vision systems for enhanced path tracking (Prakash et al., 2012). Other significant contributions include using low-pass filters to reduce light noise in color sensors (Aharari & Ueda, 2019) and developing multi-programming platforms for educational purposes (Fonseca et al., 2018). These studies offer insight into line-following robots’ technological advancements and practical applications.

The growing complexity and diverse applications of line-follower robots necessitate a comprehensive synthesis of existing research to guide future innovations. As these robots play a critical role in fields ranging from industrial automation to educational tools, a meta-analytic review is crucial for consolidating fragmented findings, identifying trends, and uncovering research gaps. This review is particularly important for researchers, engineers, and educators who seek to understand the field’s current state, leverage existing knowledge, and drive the next wave of advancements. This work aims to support the development of more efficient, adaptable, and intelligent robotic systems by providing a clear overview of the technological progress and pinpointing areas requiring further exploration.

The field of line-following robotics has experienced substantial advancements in recent years, particularly with the integration of Artificial Intelligence (AI) and machine learning (ML) techniques, which have expanded its real-world applications. These technologies have enabled line-following robots to operate with greater adaptability, precision, and autonomy, making them highly effective in dynamic environments such as automated warehouses, smart transportation systems, and industrial manufacturing lines.

This article contributes to the scientific community by providing a comprehensive overview of the advancements and challenges in line-following robotics by meta-analytical, reviewing, and analyzing the literature. This review not only differentiates itself by covering a broader scope than the previous review found it but also by offering a critical analysis that highlights gaps in the current research and suggests avenues for future exploration. The findings of this review will be invaluable for researchers, engineers, and educators looking to develop more efficient, adaptable, and intelligent line-following robots.

Materials and Methods

Literature acquisition

The literature review was conducted following the principles of the Theory of the Consolidated Meta-analytic Approach (aka TEMAC) methodology. TEMAC is structured into three distinct stages (Mariano & Santos, 2017): Stage 1: Research preparation—This stage involves defining the search strategy, including selecting keywords and databases. It also includes setting the time frame and identifying the relevant areas of knowledge to be explored.

Stage 2: Data presentation and interrelationship—In this stage, bibliometric data, such as the most prolific journals, countries, and authors in the field, are analyzed. Additionally, this stage identifies the topics most closely related to the research subject, providing insights through related keywords and concepts.

Stage 3: Detailed analysis, integrative model, and validation by evidence—The final stage involves in-depth analysis, including co-citation and coupling data. It also incorporates creating heat maps and data networks to visually represent the relationships and validate the findings through empirical evidence.

This work’s limitation is that the authors have limited funds (own funds) and, therefore, not all articles could be read, either due to the high cost or because they were not found in the university’s databases.

Data analysis

Data analysis was performed using R software with the following settings: R version 4.4.1 (2024-06-14), x86_64-pc-linux-gnu

Running under: Ubuntu 22.04.3 LTS

Matrix products: default

BLAS: /usr/lib/x86_64-linux-gnu/openblas-pthread/libblas.so.3

LAPACK: /usr/lib/x86_64-linux-gnu/openblas-pthread/libopenblasp-r0.3.20.so; LAPACK version3.10.0

Base packages: base, datasets, graphics, grDevices, methods, stats, utils

Other packages: bibliometrix 4.3.0, dplyr 1.1.4, readr 2.1.5, stringdist 0.9.12

Loaded via a namespace (and not attached): base64enc 0.1-3, bibliometrixData 0.3.0, ca 0.71.1, cellranger 1.1.0, cli 3.6.3, colorspace 2.1-1, compiler 4.4.1, crayon 1.5.3, data.table 1.15.4, digest 0.6.36, dimensionsR 0.0.3, DT 0.33, evaluate 0.24.0, fansi 1.0.6, fastmap 1.2.0, forcats 1.0.0, generics 0.1.3, ggplot2 3.5.1, ggrepel 0.9.5, glue 1.7.0, grid 4.4.1, gtable 0.3.5, hms 1.1.3, htmltools 0.5.8.1, htmlwidgets 1.6.4, httpuv 1.6.15, httr 1.4.7, igraph 2.0.3, IRdisplay 1.1, IRkernel 1.3.2, janeaustenr 1.0.0, jsonlite 1.8.8, later 1.3.2, lattice 0.22-6, lazyeval 0.2.2, lifecycle 1.0.4, magrittr 2.0.3, Matrix 1.7-0, mime 0.12, munsell 0.5.1, openalexR 1.4.0, openxlsx 4.2.6.1, parallel 4.4.1, pbdZMQ 0.3-11, pillar 1.9.0, pkgconfig 2.0.3, plotly 4.10.4, plyr 1.8.9, promises 1.3.0, pubmedR 0.0.3, purrr 1.0.2, R6 2.5.1, Rcpp 1.0.13, readxl 1.4.3, rentrez 1.2.3, repr 1.1.7, rlang 1.1.4, rscopus 0.6.6, scales 1.3.0, shiny 1.9.1, SnowballC 0.7.1, stringi 1.8.4, tibble 3.2.1, tidyr 1.3.1, tidyselect 1.2.1, tidytext 0.4.2, tokenizers 0.3.0, tools 4.4.1, tzdb 0.4.0, utf8 1.2.4, uuid 1.2-1, vctrs 0.6.5, viridisLite 0.4.2, XML 3.99-0.17, xtable 1.8-4, zip 2.3.1.

The R software was selected for its robustness, flexibility, and powerful data analysis capabilities, particularly when paired with the bibliometrix package, which facilitates bibliometric analyses such as co-citation, keyword mapping, and network clustering—critical for identifying trends and gaps in the literature on line-following robots (Aria & Cuccurullo, 2024b). This package supports advanced bibliometric and network analyses, including clustering, temporal graph generation, and conceptual mapping, while R’s seamless integration allows for automation and customization of analyses, significantly enhancing efficiency compared to proprietary tools (Aria & Cuccurullo, 2024b). Additionally, R ensures replicable results, supports a wide range of advanced analytical packages, and is freely accessible, making it an invaluable resource for researchers with limited budgets. The bibliometrix package further excels in managing large databases like Scopus and Web of Science, enabling tailored analyses that are instrumental in identifying key publications, influential authors, and emerging research trends, thereby bolstering the study’s insights and recommendations (Aria & Cuccurullo, 2024b).

The script is publicly available at GitHub: https://github.com/JohnnyEngineer/LineFollowerRobotLiteratureReview. The selection process was to have access to the article for reading (via the university network) and whether the article in any way used a line follower robot in its research.

Results and discussion

Research preparation

The following query was used to search the Scopus ®, Web of Science ® and The Lens (https://www.lens.org/) database (formerly called Patent Lens): “line follower robot” OR “line following robot”. The authors used the PRISMA 2020 flow diagram shown in Fig. 1 to transparently document the study selection process for the systematic review. This diagram shows how many records were identified, how many were excluded, and how many studies were included, ensuring clarity, reproducibility, and methodological rigor in the research (Page et al., 2021).

Figure 1 PRISMA 2020 flow diagram.

* Source: Page et al. (2021).

Data presentation and interrelationship

In Table 1, it is the number of documents found on each plataform.

Table 1 Search results across databases.

Database	Records found	
Scopus	267	
Web of science	111	
Patent lens	29	
Total after deduplication	287	

In Table 2 shows the percentage of missing data attribute. The quality of the data is visible.

Table 2 Data quality assessment.

Data type	Missing percentage	
Abstract, authors’ names, references, type of document, periodicals, year, title, citations	0%	
Language	4.18%	
Affiliation	9.76%	
DOI	26.13%	
Keywords	20.21%	
Author correspondents	34.49%	
Keyword plus data	52.26%	
Scientific categories	61.32%	

In Table 3 shows the timeline of production over the years for the documents.

Table 3 Dataset overview.

Category	Value	
Time period covered	2001 to 2024	
Total documents	287	
Unique journals, books, and sources	246	
Annual growth rate	12.16%	
Average document age	5.95 years	
Average citations per document	3.666	
Total references cited	1,498	

In Table 4 shows the production of research based on the types of research documents.

Table 4 Distribution of document types.

Document type	Count	
Conference papers	107	
Proceedings papers	88	
Articles	62	
Conference reviews	13	
Book chapters	4	
Journal articles	9	
Books	1	
Systematic reviews	1	

In Table 5 shows the production of research based on the authorship and collaboration of the authors documents.

Table 5 Authorship and collaboration.

Metric	Value	
Total authors	864	
Author appearances	1,028	
Single-authored documents	14	
Average documents per author	0.332	
Average co-authors per document	3.58	

In Table 6 shows the production of research based on the author’s countries.

Table 6 Top countries in line-following robotics research.

Country	Articles (% of base)	Citations (Avg. per article)	
India	44 (23.40%)	97 (2.205)	
Brazil	14 (7.45%)	–	
Indonesia	14 (7.45%)	72 (5.143)	
Bangladesh	11 (5.85%)	52 (4.727)	
China	9 (4.79%)	42 (4.667)	
Malaysia	9 (4.79%)	37 (4.111)	
Poland	8 (4.26%)	–	
Turkey	8 (4.26%)	59 (7.375)	
USA	6 (3.19%)	37 (6.167)	
Iran	5 (2.66%)	86 (17.200)	
Canada	–	30 (15.000)	
Mexico	–	26 (13.000)	

In the Table 7 shows the most frequent publications sources.

Table 7 Most frequent publication sources.

Source titles	
AIP (American Institute of Physics) Conference Proceedings	
Advances in Intelligent Systems and Computing	
Lecture Notes in Computer Science	
Latin American Robotics Symposium	
Brazilian Robotics Symposium	
CEUR Workshop Proceedings	
International Journal of Engineering and Advanced Technology	
IEEE International Conference on Smart Computing	
Journal of Engineering and Applied Sciences (ARPN)	

Table 8 shows some columns. The Total Citations (TC) column displays how many times each article has been cited since publication, indicating its impact and recognition within the academic community (Aria & Cuccurullo, 2024a). The Citations per Year (TCperYear) normalizes the total citations by the number of years since the article’s publication, offering a measure of the article’s average annual impact (Aria & Cuccurullo, 2024a). The Normalized Citation Count (NTC) adjusts the total citations by factors such as the publication venue’s citation influence and the overall citation environment (Aria & Cuccurullo, 2024a).

Table 8 Top manuscripts per citations.

Paper	TC	TCperYear	NTC	
Pakdaman & Sanaatiyan (2009)	34	2.12	2.00	
Lee et al. (2010)	32	2.13	2.25	
Bajestani & Vosoughinia (2010)	32	2.13	2.25	
Dupuis & Parizeau (2006)	28	1.47	1.00	
Saadatmand et al. (2020)	26	5.20	4.86	
Thirumurugan et al. (2010)	25	1.67	1.75	
Almasri, Elleithy & Alajlan (2015)	22	2.44	3.93	
Latif et al. (2020)	22	4.40	4.11	
Engin & Engin (2012)	21	1.62	3.71	
Note:

TC: Total Citations; TCperYear: Citations per Year; NTC: Normalized Citation Count.

Table 8 suggests Pakdaman & Sanaatiyan (2009), Lee et al. (2010), and Bajestani & Vosoughinia (2010) have the highest total citations, reflecting their long-standing influence. However, more recent articles like Saadatmand et al. (2020) are gaining traction quickly, as evidenced by its high TCperYear and NTC scores. The high TCperYear and NTC values for recent articles like this Saadatmand et al. (2020) suggest emerging research trends, indicating that these topics may be of increasing importance and relevance in the field. Older articles like Dupuis & Parizeau (2006) and Almasri, Elleithy & Alajlan (2015) continue to be cited, indicating their sustained relevance. In contrast, newer articles are quickly making an impact, which might suggest shifts in research focus or the introduction of innovative ideas (Aria & Cuccurullo, 2024a).

Pakdaman & Sanaatiyan (2009) discuss designing, implementing, and testing a line follower robot named “TABAR,” developed for a line follower robots competition. The robot is designed to follow a pre-defined path autonomously, typically a black line on a white surface, using a series of infrared (IR) sensors. The authors present a comprehensive guide on the robot’s structure, the electronic components used, programming, and the challenges encountered during its development. The primary contributions of the article include a detailed description of the robot’s hardware architecture, the programming logic used to interpret sensor data and control the robot’s movement, and practical tips on sensor placement, circuit design, and programming. The robot uses an array of eight IR sensors to detect the line’s position, and an Atmel AVR microcontroller processes the sensor data and controls the robot’s motors. The robot successfully follows lines with various configurations, and the authors note that its performance is influenced by the quality of the sensors, the precision of the programming, and the overall weight and design of the robot. They emphasize the importance of further research and development to enhance the robot’s capabilities.

Lee et al. (2010) present a project-based laboratory designed to teach embedded system design, focusing on developing a line-following robot. According to the authors, the laboratory was developed with support from Microchip Inc., allowing students to access microcontrollers and C-compilers at no cost. The primary contribution of their work is creating an educational environment where students learn through hands-on experience, reinforcing theoretical knowledge with practical applications. Key contributions of the project-based approach include practical embedded system design, industry support, and competition-based learning. According to Lee et al. (2010), the lab’s methodology is project-based, with students progressing through exercises that build their skills in embedded system design. The article suggests the lab successfully equips students with the skills necessary to design and build a line-following robot, culminating in a racing contest. The authors identify several limitations of the lab, including cost constraints, learning curve, and resource intensity (Lee et al., 2010). The project-based laboratory provides a robust framework for teaching embedded system design, effectively motivating students and enhancing their learning outcomes. However, the article suggests that cost reduction and additional support are needed to make the lab accessible to a broader range of students.

Bajestani & Vosoughinia (2010) discuss the design, construction, and testing of a line follower robot developed by undergraduate students at the Islamic Azad University, Mashhad branch. Key contributions include the robot’s sensor configuration, advanced line tracking capabilities, and cost-effective design. The methodology involves the sensor system, comparator circuit, microcontroller, motor control, power supply, and chassis design (Bajestani & Vosoughinia, 2010). The robot successfully demonstrated the ability to follow various line configurations, but limitations include noise sensitivity, mechanical constraints, and power consumption (Bajestani & Vosoughinia, 2010). The article suggests that while the robot performs well, there are areas for further improvement, particularly in reducing noise sensitivity and enhancing chassis durability, providing a foundation for future work in robotics education (Bajestani & Vosoughinia, 2010).

Dupuis & Parizeau (2006) present a framework for evolving a vision-based mobile robot controller using genetic programming. The researchers aimed to develop a controller capable of guiding a robot to follow a line using visual input without manual programming. Their key contributions include integrating high-resolution visual data, genetic programming for controller evolution, a detailed simulation environment, and dynamic environment configuration. The study successfully evolved controllers capable of following a line, achieving a fitness level of approximately 70%. However, limitations were identified in the fitness function and in transferring evolved behaviors to a physical robot. The research demonstrates the feasibility of using genetic programming to evolve vision-based robot controllers in a simulated environment. Still, it highlights the need for more sophisticated simulation models and refined fitness evaluation methods for real-world applicability. Future work should focus on improving the simulator’s fidelity, developing more effective fitness functions, and exploring using state primitives to enhance the controller’s ability to exploit past experiences (Dupuis & Parizeau, 2006).

Saadatmand et al. (2020) discuss the development of an autonomous line follower robot using a Q-learning controller enhanced by simulated annealing and their key contributions include integrating Q-learning with simulated annealing, presenting a detailed mathematical model for the robot, and validating the controller’s effectiveness through simulations and experiments. The study’s key results show improved performance, learning efficiency, and robustness in complex environments. Limitations of the research include the complexity of implementation, training time, and real-world uncertainties. Saadatmand et al. (2020) conclude that the SA-based Q-learning controller provides a significant improvement and suggests its potential application to other robotic systems while acknowledging the need for further enhancements in real-world environments.

Thirumurugan et al. (2010) discuss a Line Following Robot designed for library inventory management and their key contributions include its application in library management, integrated barcode scanning, and automation of routine tasks. The methodology inside their article covers hardware design, control systems, and barcode scanning with database integration. The prototype robot successfully located books and tracked shelf arrangement but has limitations in database capacity, manual input requirement, and single scanner use (Thirumurugan et al., 2010). The authors conclude that the Line Following Robot prototype enhances library management efficiency and suggests future improvements such as wireless communication with the library’s Online Public Access Catalog (OPAC) and integration with book loan systems (Thirumurugan et al., 2010).

Almasri, Elleithy & Alajlan (2015) present a study on a sensor fusion-based model for mobile robot navigation. Their model integrates sensors using fuzzy logic to detect and avoid obstacles while following a path and their key contributions include fuzzy logic-based sensor fusion, simulation, and real-time testing. The methodology of their article involves robot and environment modeling, fuzzy logic system design, fuzzy rule design, and simulation and real-time implementation. Besides that, other key results include improved obstacle detection, effective navigation, and reduced travel distance. The limitations include sensor limitations, the complexity of fuzzy logic design, and real-world variability (Almasri, Elleithy & Alajlan, 2015). The authors conclude that the proposed model enhances collision-free navigation and suggests future work in optimizing the fuzzy logic system and exploring other sensor fusion techniques.

According to Latif et al. (2020), the design and testing of a line follower robot using the ATMega32A microcontroller. Their key contributions include the development of a simple educational robot, showcasing the use of the ATMega32A microcontroller, and integrating mechanical, electronic, and software components. The robot successfully followed a line at lower speeds but faced limitations at higher speeds due to sensor sensitivity issues. The authors suggest further improvements for enhanced functionality at higher speeds and serve as a valuable resource for educational robotics (Latif et al., 2020).

Engin & Engin (2012) suggest a design, implementation, and path planning of a line follower robot using a dynamic PID control algorithm. The robot was designed to navigate autonomously along a predefined path using differential drive locomotion, controlled by a Stellaris microcontroller. Conforming to Engin & Engin (2012), one of the key contributions of the article includes the development of a dynamic PID control algorithm, the integration of sensors and the microcontroller, and its educational application. Their methodology involved robot design and structure, sensor placement, PID control algorithm, and experimental setup. Their results include improved performance with dynamic PID and reduced oscillation and error. One of the limitations includes the manual tuning of PID parameters and sensitivity to environmental conditions. The authors suggest that the dynamic PID control algorithm significantly enhances the performance of line-following robots and holds potential for educational applications (Engin & Engin, 2012).

A graph was generated from co-words, as shown in Fig. 2. This co-word analysis graph, generated by Bibliometrix, visualizes the relationships between keywords that frequently occur together in the literature on line-following robots and related topics (Aria & Cuccurullo, 2024a). This co-word analysis reveals the central themes and emerging trends in the research on line-following robots. Key areas like robotics, control systems (e.g., PID algorithms), and hardware implementations (e.g., Arduino) are well-established in the literature. Additionally, the graph indicates growing interest in simulation techniques like digital twins and co-simulation, which could represent future directions for research in this field. This visualization helps to identify both well-trodden paths and emerging areas that might warrant further exploration.

Figure 2 Co-words.

Nodes (Circles)—Size: The size of each node represents the frequency of a keyword’s occurrence in the analyzed documents. Larger nodes indicate that the keyword appears more frequently in the documents (Aria & Cuccurullo, 2024a).

Nodes (Circles)—Color: Different colors represent different clusters of related keywords. Keywords within the same cluster tend to co-occur more often with each other in the literature, suggesting they are related topics or are part of a common research theme (Aria & Cuccurullo, 2024a).

Edges (Lines)—Thickness: The thickness of the lines connecting the nodes represents the strength of co-occurrence between two keywords. Thicker lines indicate a stronger relationship, meaning the two keywords frequently appear together in the same documents (Aria & Cuccurullo, 2024a).

Edges (Lines)—Length: The length of the edges generally reflects the degree of association. Shorter lines indicate a closer association between keywords, while longer lines suggest a weaker relationship (Aria & Cuccurullo, 2024a).

Based on the above concept about co-word analysis, it is possible to conclude the following facts: Robotics Cluster (Red): The largest node is “robotics,” indicating it is a central theme in the literature. This cluster is connected to other important keywords like “mobile robot,” “line follower,” and “obstacle detection,” suggesting these are core topics within this field of study.

Arduino and Microcontroller Cluster (Blue): Another significant cluster revolves around “Arduino” and “microcontroller.” This indicates that much of the literature focuses on the hardware aspect of line-following robots, particularly the use of Arduino microcontrollers in building and programming these robots.

PID and Control Algorithms Cluster (Purple): The presence of keywords like “PID” and “line following” in a closely related cluster indicates a strong focus on control strategies in the literature. This cluster likely includes research on improving the accuracy and performance of line-following robots through various control algorithms.

Digital Twin and Simulation Cluster (Pink): There is a smaller but distinct cluster related to “digital twin” and “co-simulation,” indicating emerging research in simulation and virtual modeling of robotic systems, which might be a newer trend in the literature.

Isolated Clusters: Some keywords, such as those related to specific methodologies or tools (“co-simulation,” “image processing”), are more isolated or connected to fewer other keywords. This suggests these topics might be specialized or less integrated into the broader themes of the field.

The authors of this article used these tips to carry out research in the article on simulation of line-following robots (Brigido, Maximo & Parente de Oliveira, 2024).

Figure 3 is often used to explore relationships between categories of qualitative variables in a reduced multidimensional space (Aria & Cuccurullo, 2024a). Dim 1 explains 62.41% of the variation, while Dim 2 explains 29.42%. Together, these two dimensions cover a significant portion of the variability in the data, facilitating interpretation (Aria & Cuccurullo, 2024a). Points that are close to each other represent categories that share similar characteristics or that are frequently associated with the data (Aria & Cuccurullo, 2024a). Distant dots indicate categories that are distinct or have little in common (Aria & Cuccurullo, 2024a). It is possible to deduce the points group on the right—dim 1 Positive (0 to 1.0)—are the categories “line follower,” “robot,” and “PID,” which are close to the “Buildometrix” logo. This suggests that these categories have a stronger relationship with each other or are often associated with conceptual analysis (Aria & Cuccurullo, 2024a). The group points on the left—dim 1 Negative (−1.5 to −0.5)—are the categories “line follower robot” and “Arduino”, suggesting that these categories are distinct from the group on the right. The group points above—dim 2 Positive (0 to 1.0)—are the categories “line following robot” and “robotics” grouped in the upper right corner, suggesting an association between them that is different from the categories further down (Aria & Cuccurullo, 2024a). The group points on the bottom center left is the category “microcontroller,” which is a little bit separated from the other groups, which may indicate that it has different characteristics or is a less associated concept compared to the others (Aria & Cuccurullo, 2024a). It’s possible to conclude that the X axis is the dimension that separates concepts related to “line follower,” “robot,” and “PID” (on the right) from concepts such as “Arduino” and “line follower robot” (on the left) and the Y axis separates concepts more related to “robotics” (above) from other concepts such as “Arduino” (below) (Aria & Cuccurullo, 2024a).

Figure 3 Conceptual structure map—multiple correspondence analysis (MCA).

Figure 4 is used to visualize the formation of hierarchical clusters and the similar relationships between different items or concepts. According to Aria & Cuccurullo (2024a), the Y axis represents the distance or dissimilarity between the clusters. The greater the height of the node where two branches join, the greater the dissimilarity between the clusters or concepts. Each label on the X-axis represents a concept or item that is being grouped. In your graph, concepts include terms such as “line following robot”, “robotics”, “pid”, “line follower,” “robot”, “line follower robot”, “Arduino” and “microcontroller”. The “line following robot” and “robotics” are grouped together at a low level, suggesting that these two concepts are very similar. If two concepts or clusters are joined at a very high level in the dendrogram, this indicates that they are more different from each other (Aria & Cuccurullo, 2024a). For example, the group containing “line follower”, “robot”, “line follower robot”, “Arduino” and “microcontroller” is only joined to the group of “line following robot” and “robotics” in one high-level, indicating greater dissimilarity. Figure 5 shows the distribution of documents with the largest contributions in a conceptual space, using the two main dimensions (Dim 1 and Dim 2) (Aria & Cuccurullo, 2024a). These two dimensions represent the main directions that explain most of the variability in the data. Dim 1 explains 62.41% of the variation, and Dim 2 explains 29.42%. Together, these dimensions cover approximately 92% of the variability, making the graph a good representation of the data. Documents to the right of the X-axis (for example, the Bibliometrix logo and the “cluster2” document) are grouped together, suggesting that these documents are related in some way, probably more technological or engineering-focused themes. Conforming to Aria & Cuccurullo (2024a), documents that are close to each other in the graph share themes, topics, or methodological approaches and documents that are distant in the graph indicate a dissimilarity in terms of content or approach. It is possible to conclude the existence of five clusters: Cluster 1 (Red): This is a smaller and more isolated cluster, indicating that the documents here have less relationship with the other groups.

Cluster 2 (Blue): The document is largely isolated in the lower right quadrant, suggesting a unique thematic focus or distinct methodology.

Cluster 3 (Green): Documents such as “parihar s, 2023” and “amorim j, 2022” are close together and grouped in a cluster, indicating a similarity in their themes, which may be related to technological innovations and robotics education.

Cluster 4 (Purple): This group contains documents such as “alias m, 2017” and “oprea m, 2020”, which appear to be related to information technology and computer science education topics.

Cluster 5 (Orange): This cluster contains documents such as “shah m, 2017” and “shah m, 2018”, suggesting a continuity or thematic connection between these works, possibly related to the transformation in engineering education.

Figure 4 Conceptual structure map—hierarchical clustering analysis (HCA).

Figure 5 Conceptual structure map—document factor map.

Amorim et al. (2023) discuss the optimization of a line follower robot’s performance through control tuning and digital signal processing and their key contributions include innovative control tuning, application of digital filters, hardware and software enhancements. The study’s methodology involves a differential robot concept, PID control implementation, digital filter application, and hardware configuration. Other key results of the article include improved performance with PID and digital filters, noise reduction in sensor readings, and enhanced stability and speed (Amorim et al., 2023). Limitations include tuning complexity, sensor dependence, and sampling delay. The authors suggest significant enhancements in robot performance (Amorim et al., 2023).

Oprea (2020) discusses the integration of robotics projects into pre-university education. It emphasizes the importance of preparing students for future careers in areas like smart home development, human-robot interaction, and augmented reality. The study highlights the use of the Arduino platform to implement robotics projects such as a line follower robot and an obstacle-avoiding robot, demonstrating how these projects can enhance STEM (Science, Technology, Engineering, and Mathematics) education. Some of the key contributions of the article include the promotion of STEM education, the practical implementation of robotics projects and the comparative analysis. To do this, developing robotics projects, implementing educational programs, and collecting and analyzing data were used. Some of the results include increased student engagement, enhanced learning outcomes, and positive feedback from educators (Oprea, 2020). However, the study also identifies limitations related to resource availability, teacher training, and time constraints (Oprea, 2020). Oprea (2020) suggests the integrating robotics projects into pre-university education is highly beneficial for fostering interest in STEM fields and equipping students with the skills needed for future technological careers. It suggests that for broader implementation, schools need to invest in necessary resources and provide adequate training for teachers.

Shah, Rawal & Dalwadi (2017) present the design and implementation process of a high-performance line follower robot, developed for participation in the ROBOCON 2016 international robotics competition. The study focuses on optimizing sensor selection, signal processing, and sensor arrangement to enhance the robot’s ability to precisely follow lines, especially in complex environments with varied backgrounds and sharp turns. Some of the key contributions include optimized sensor selection and configuration, a comparison of signal processing methods, and an exploration of sensor arrangement for complex paths. Their methodology involves sensor selection, signal processing exploration, sensor arrangement, and implementation and testing. The study’s results highlight high accuracy in line detection, effective handling of complex paths, and improved robot performance. The authors acknowledge limitations such as environmental sensitivity, higher component costs, and the complexity of implementation (Shah, Rawal & Dalwadi, 2017). They conclude that careful sensor selection and arrangement, combined with appropriate signal processing, can significantly enhance line-following robot performance in competitive environments (Shah, Rawal & Dalwadi, 2017). The research highlights possibilities of research in reducing sensitivity to environmental changes, exploring cost-effective solutions, and automating the calibration process to make the design more accessible.

All other articles described in Fig. 5 were not accessible for this review.

Detailed analysis, integrative model, and validation by evidence

According to Vogel & Güttel (2012), co-citation provides insights into the most influential authors, the degree of collaboration, and the emerging trends in the research domain. Coupling is carried out taking into account only the last 3 years of the research. Bibliographic coupling occurs when two or more authors cite the same references in their work, indicating that they may be working on related topics or drawing from similar research foundations (Vogel & Güttel, 2012). It is only considered the last 3 years of publication (Vogel & Güttel, 2012).

Figure 6 is the co-citation analysis network. The largest node in the network is labeled “anonymous,” which likely represents references to works where authorship is not specified or is grouped under collective authorship (e.g., corporate reports, standards, or guidelines). This suggests that some foundational or widely used resources in the field may not be directly attributed to specific individuals. Other significant nodes include authors such as Pakdaman & Sanaatiyan (2009), Engin & Engin (2012), and Palmieri, Bernardeschi & Masci (2018), indicating these individuals are frequently cited together and have made influential contributions to the field of line-following robots. The network shows distinct clusters of authors, each likely representing a particular research focus or collaboration network within the broader field of line-following robots (Aria & Cuccurullo, 2024a). For instance: Pakdaman & Sanaatiyan (2009) and Engin & Engin (2012) appear in close proximity, suggesting a shared research focus or frequent collaboration.

Bernardeschi, Domenici & Palmieri (2019) and Palmieri, Bernardeschi & Masci (2018) are other notable authors who, despite being less central than Pakdaman or Engin, form their own influential clusters, possibly focusing on specific subtopics within the field, such as safety-critical systems or formal methods in robotics.

Figure 6 Co-citation authors.

Figure 6 shows some authors, such as Hasan, Abdullah-Al-Nahid & Al Mamun (2012) and Hasan et al. (2013), appear more isolated in the network. This could indicate specialized research areas that are less frequently co-cited with the main clusters, or it could suggest that their work, while influential, is cited in different contexts than the central themes of the network (Aria & Cuccurullo, 2024a). The presence of smaller, less connected clusters (such as those around Kaiser et al. (2014)) indicates emerging research groups or niche areas within the field of line-following robots. These clusters could represent innovative or interdisciplinary approaches that are beginning to gain traction in the literature (Aria & Cuccurullo, 2024a; Vogel & Güttel, 2012).

The conclusion is that the prominent position of Pakdaman & Sanaatiyan (2009) and Engin & Engin (2012) suggests that their research areas are highly influential and could serve as a foundation for further studies. In contrast, the isolated nodes represent opportunities for cross-disciplinary research, where integrating insights from these authors with the broader field could lead to new innovations.

Palmieri, Bernardeschi & Masci (2018) discuss the modeling and the simulating attacks on cyber-physical systems (CPS), focusing on the impact of attacks on sensor and actuator behavior. It uses formal methods and the INTO-CPS framework for co-simulation, demonstrating the methodology with a line follower robot case study. The results show how attacks can disrupt system behavior (Palmieri, Bernardeschi & Masci, 2018). However, the study highlights complexities in modeling, scalability challenges, and the need for verification of system properties under attack (Palmieri, Bernardeschi & Masci, 2018). Overall, the research suggests that the proposed methodology is effective in understanding CPS vulnerabilities (Palmieri, Bernardeschi & Masci, 2018).

Bernardeschi, Domenici & Palmieri (2019) discuss a methodology for modeling and simulating security attacks on cyber-physical systems (CPS). The study focuses on the impact of attacks on system behavior, using a line follower robot as a case study. Some of the key contributions include formal modeling of attacks, the use of the INTO-CPS framework for co-simulation, and practical application to real-world scenarios. The methodology was a PVS environment, co-simulation with INTO-CPS, and modeling of attacks (Bernardeschi, Domenici & Palmieri, 2019). Some of the results show the impact of attacks on system behavior and provide detailed execution traces for analysis. Suggestions for future research include the complexity of modeling, scalability, and lack of verification. The study concludes that the proposed methodology effectively reveals vulnerabilities and the impact of attacks. Future work is suggested to extend the models to include more sophisticated attack scenarios and explore verification techniques (Bernardeschi, Domenici & Palmieri, 2019).

Hasan, Abdullah-Al-Nahid & Al Mamun (2012) presented the design of a simple but effective robot that follows a line using basic electronic components. It used a feedback mechanism and can handle tight curves. The key contributions include cost-effective design, a simple feedback mechanism, and the ability to follow a line on different surfaces (Hasan, Abdullah-Al-Nahid & Al Mamun, 2012). The robot’s methodology involves sensor setup, a comparator and logic circuit, and motor control, and their results show that the robot can follow a line at a moderate speed and navigate tight curves (Hasan, Abdullah-Al-Nahid & Al Mamun, 2012). However, it has limitations in handling complex paths and environments and lacks sophistication for more advanced applications (Hasan, Abdullah-Al-Nahid & Al Mamun, 2012). The study suggests further modifications to enhance the robot’s capabilities (Hasan, Abdullah-Al-Nahid & Al Mamun, 2012).

Hasan et al. (2013) suggest the Multiple Source Multiple Destination Robot (MDR-1), an advanced line follower robot capable of autonomously choosing and following a line among multiple colored lines while avoiding obstacles. One of the key contributions includes color detection, obstacle avoidance, and enhanced navigation capability. Their methodology was a sensor setup, color separation, and control logic. The robot successfully demonstrated its abilities in various scenarios but has limitations related to the complexity of implementation, sensor sensitivity, and processing power (Hasan et al., 2013). The conclusion was that the MDR-1 represents a significant advancement in autonomous robotics, with potential applications in industrial automation and dynamic environments (Hasan et al., 2013).

Kaiser et al. (2014) explored the design, fabrication, and performance evaluation of a line-follower robot, focusing on the accuracy of the robot’s line-following capabilities using data acquisition methods and their key contributions include the fabrication of a line-follower robot, accuracy measurement using data acquisition, and performance analysis. The methods involved component selection and fabrication, sensor and motor control, data acquisition, and simulation and testing. The study’s key results include the accuracy of line following, motor performance, and noise in sensor data. The study identifies several limitations, including motor driver constraints, sensor noise, and power consumption. In conclusion, the research suggests areas for improvement, particularly in terms of motor control and sensor accuracy, and highlights the importance of sensor quality and motor driver capabilities in the overall design.

Figure 7 shows a co-citation analysis of sources which is a method used to understand how often different journals, conferences, and other publication sources are cited together in academic literature (Aria & Cuccurullo, 2024a; Vogel & Güttel, 2012). It is possible to conclude the red cluster is the largest and includes prominent sources like “Procedia Computer Science”, “IEEE Transactions on Education” and “International Conference on Computer and Electrical Engineering”. This cluster seems to focus on educational aspects of robotics, engineering education, and the application of robotics in educational settings. The blue cluster includes sources such as “Robotics and Autonomous Systems”, “Applied Soft Computing” and “IEEE Transactions on Control Systems Technology”. These sources are typically associated with the technical and engineering aspects of robotics, including control systems, soft computing applications, and autonomous systems. The green cluster includes sources like “Lecture Notes in Computer Science”, “IEEE Transactions on Software Engineering” and “9th International Modelica Conference”. These sources are more focused on software engineering, computational methods, and formal modeling in robotics and automation. The purple cluster includes sources such as “Concurrency and Computation: Practice and Experience” and “Communicating Process Architectures”. These sources likely focus on parallel computing, concurrent systems, and the application of these concepts in robotics and automation.

Figure 7 Co-citation sources.

The size of the nodes represents the frequency of co-citation, with larger nodes indicating more frequently co-cited sources (Aria & Cuccurullo, 2024a). For example, the “International Conference on Computer and Electrical Engineering” in the red cluster is highly co-cited, suggesting its central role in this area of research. “Lecture Notes in Computer Science” in the green cluster also appears prominently, indicating its widespread influence in computational and software aspects of robotics. The edges (lines) between nodes represent the strength of co-citation relationships (Aria & Cuccurullo, 2024a). Thicker and shorter edges indicate stronger co-citation links, meaning the sources are frequently cited together (Aria & Cuccurullo, 2024a). So, there are notable connections between different clusters, such as between the red and blue clusters, indicating that research in educational robotics is often connected with technical and engineering research in robotics. This suggests interdisciplinary research efforts that bridge education and practical application in robotics. Some sources, such as those in the purple cluster, are more isolated, indicating that they specialize in niche areas of robotics and automation that do not overlap extensively with mainstream robotics research.

In their work, Fonseca et al. (2018) proposed a new method to upgrade the Pololu 3Pi robot, transforming it into a multi-programming platform. The main objective is to address the inconvenience of having to upload a new program for each trial when developing and testing line-following robots, particularly in educational settings. The key contributions of this approach include introducing multi-programming capability, enhancing efficiency in robotics education, and optimizing memory usage. The methodology of the study involves developing the 3piMulti tool, structuring the program, and testing memory and performance. The significant results show time savings, improved memory efficiency, and an enhanced user experience. However, the study identifies limitations such as restrictions on reserved words and the absence of support for previous program storage. The research concludes that the multi-programming tool substantially improves the functionality of the Pololu 3Pi, especially in educational environments where time efficiency is crucial. This work provides a valuable contribution to the field of educational robotics, offering a practical solution to common challenges encountered when working with limited hardware platforms like the Pololu 3Pi.

The analysis in Fig. 8 shows the coupling between authors. Each node represents an individual author, with the size of the node indicating the strength of their coupling. Larger nodes suggest a higher degree of coupling with others in the network, indicating a central role in the research community. The lines connecting the nodes represent the strength of coupling between the authors, with thicker lines indicating a stronger connection. The length of the lines can also indicate the closeness of the relationship, with shorter lines representing stronger connections (Aria & Cuccurullo, 2024a).

Figure 8 Coupling authors.

Different colors indicate clusters of authors who are more closely coupled with each other. These clusters represent groups of researchers who are likely working on similar topics or themes within the broader field. The red cluster includes authors such as “Benavides-Robles M”, “Valencia-Rivera G”, “Cruz-Duarte J”, “Amaya I” and “Ortiz-Dayliss J”. The dense connections between these authors suggest a strong research collaboration or that they frequently reference similar sources, indicating they are likely working in closely related areas of research. The blue cluster includes authors like “Alhuwaimel S”, “Farh R”, “Al J K” and “Quasim M”. Similar to the red cluster, these authors are closely connected, indicating they work on related topics or share a common research base (Aria & Cuccurullo, 2024a; Vogel & Güttel, 2012). “Bernardeschi C” appears as an isolated node with no direct connections to other authors in the graph. This suggests that this author’s work does not frequently share references with the others represented in this graph or they work in a more niche area that is not directly connected to the clusters shown.

Benavides-Robles et al. (2024) made a systematic review of the Robotic Mobile Fulfillment System (RMFS), focusing on the role of Line Follower Robots (LFRs) in automated warehouses. The study aimed to synthesize existing research on RMFS, identify gaps in the literature. Benavides-Robles et al. (2024) analyze 76 relevant articles and emphasize the need for more research on pod allocation, replenishment strategies, and energy-saving approaches within RMFS. They highlight the fragmentation of RMFS literature, the lack of research on pod allocation and energy-saving strategies, and the variability of simulation assumptions. The article proposes a clearer classification scheme for RMFS research and calls for more comprehensive studies that address the entire RMFS, including the less-explored areas like pod allocation and energy management. This article was co-authored by Benavides-Robles, Valencia-Rivera, Cruz-Duarte, Amaya and Ortiz-Bayliss (Benavides-Robles et al., 2024).

Farkh et al. (2021) presented a deep-learning approach to improving the motion control system of a line-following robot. The study demonstrated the use of a multilayered feedforward neural network trained via backpropagation to control the robot, showcasing improved navigation accuracy and robustness to environmental variations. The authors also highlighted the low-cost implementation on an Arduino platform and suggested the potential for future work in integrating more advanced deep-learning techniques. This article was co-authored by Farkh, Al Jaloud, Alhuwaimel, Quasim, and Ksouri (Farkh et al., 2021).

According to Bernardeschi, Domenici & Palmieri (2020), there is a methodology for modeling security attacks on CPS (aka vulnerability management) that focuses on sensors and actuators and their key contributions include an integrated co-simulation framework and formal modeling of attacks using the prototype verification system (PVS). Their methods involve system and attack modeling, co-simulation, and formal verification. Bernardeschi, Domenici & Palmieri (2020) suggests the effectiveness of the approach in identifying system vulnerabilities and confirming system properties. Some of the limitations include the scope of attacks, complexity of modeling, and computational overhead (Bernardeschi, Domenici & Palmieri, 2020). Bernardeschi, Domenici & Palmieri (2020) conclude that the approach provides a robust framework for analyzing the security of CPS and is particularly relevant for safety-critical systems. Some gaps were identified for future work, such as expanding the methodology to cover a wider range of attack types and exploring strategies for optimizing the co-simulation process (Bernardeschi, Domenici & Palmieri, 2020).

This historical direct citation network provides a clear visualization of the development of research in the field, showing how foundational articles have influenced subsequent research and how new ideas and themes are emerging over time (Aria & Cuccurullo, 2024a). Each node represents a specific publication. The size of the node typically indicates the article’s impact, often measured by the number of citations it has received. Larger nodes are more influential in the field (Aria & Cuccurullo, 2024a). The color of the nodes usually corresponds to different clusters or themes within the network, suggesting that the publications contribute to different areas or subfields of the research topic (Aria & Cuccurullo, 2024a). The lines connecting the nodes represent direct citations from one publication to another. These connections illustrate how ideas and research findings have been built upon over time (Aria & Cuccurullo, 2024a).

Figure 9 show the historical influence of articles published over time. With this image, it is necessary to analyze the articles Mkaouar et al. (2019), Yildiz et al. (2020), Gawlicki & Jankowski (2021), Gosim et al. (2012).

Figure 9 Historical citation analysis the lens.

Figure 10 shows early influential articles like Pakdaman & Sanaatiyan (2009) and Lee et al. (2010), which appear as larger nodes. These articles have been foundational in the field and have influenced subsequent research. The connections from these early articles to later publications indicate that they have served as key references for ongoing research. As time progresses, we see the network expanding with more nodes and connections, particularly between 2015 and 2018. This period likely represents a time of growth in the research area, with new articles building on the foundational work of earlier years. More recent articles, such as those by Bernardeschi, Domenici & Palmieri (2020) and Palmieri, Bernardeschi & Masci (2018), are positioned towards the right side of the timeline, indicating their recency and ongoing influence in the field. The clusters of nodes in the 2020–2023 period suggest emerging areas of research, with new connections forming between these recent articles and older works. This reflects the continued evolution and diversification of the research field. The different colored clusters indicate that the field has diversified into several research themes or subfields. For example, the pink and green clusters in recent years suggest new or distinct research directions that are beginning to influence the field. The yellow and red nodes, such as those representing Prakash et al. (2012), may represent specialized topics that have developed alongside the more mainstream research. Certain nodes act as bridges between different clusters, indicating articles that have connected disparate areas of research. For example, Aharari & Ueda (2019) might be linking older foundational work with newer, more specialized research.

Figure 10 Historical citation analysis—web of science.

Mkaouar et al. (2019) presented a formal approach to model-based software engineering for safety-critical real-time systems using Architecture Analysis and Design Language (AADL). Mkaouar et al. (2019) introduce a method to integrate formal verification into the AADL model-driven development process and demonstrates its applicability with case studies. The key results include successful verification of complex systems and scalability, but the study also acknowledges limitations related to the complexity of formal methods, scalability issues, and the limited scope of the AADL subset supported by the toolchain (Mkaouar et al., 2019). Overall, the research emphasizes the value of integrating formal verification into the AADL model-driven development process for enhancing the reliability and safety of real-time systems (Mkaouar et al., 2019).

Yildiz et al. (2020) discusses the implementation of a sliding mode control (SMC) strategy for a line-following robot to enhance its path-tracking accuracy and stability. The key contributions include the development of non-chattering SMC, comparison with PID control, and experimental validation. Their methods involve mathematical modeling, the design of a sliding mode controller, and implementation/testing. Yildiz et al. (2020) show superior path tracking, energy efficiency, and robustness to disturbances. Some of the limitations include the complexity of SMC implementation and dependence on sensor accuracy (Yildiz et al., 2020). The article suggests that non-chattering SMC provides a significant improvement in the performance of line-following robots, offering a viable alternative to traditional PID control for mobile robot tasks (Yildiz et al., 2020).

Gawlicki & Jankowski (2021) showed a novel approach for identifying trajectories of moving loads using multicriterial optimization. They used a framework that combines mechanical response fitting and geometric regularity criteria to ensure physically meaningful results. The method was experimentally verified and successfully identified trajectories with reduced error and increased geometric regularity compared to traditional methods (Gawlicki & Jankowski, 2021), However, the complexity of the optimization process and dependence on sensor accuracy are potential limitations (Gawlicki & Jankowski, 2021). The study suggests that the proposed approach is effective and applicable to real-world scenarios, with the potential for future extensions to more complex structures and different load types (Gawlicki & Jankowski, 2021).

Gosim et al. (2012) discuss the development of a fully automated system that integrates a line-following robot with an ABB industrial robot manipulator for material handling tasks in an industrial setting. It presents the integration process, synchronization methods, system components, and experimental results. The study highlights the successful pick-and-place operations and the system’s limitations, concluding that the integrated robotic system offers efficient material handling but has some adaptability challenges.

Prakash et al. (2012) presented the design and implementation of a line-following robot using computer vision technology. Key contributions include integrating computer vision for improved line detection and tracking, real-time image processing, and enhanced navigation capabilities. The methodology involves system design with a camera, image processing algorithm, and hardware integration. Key results include improved line tracking accuracy, adaptability to complex paths, and real-time performance. Limitations include dependence on external processing, environmental sensitivity, and processing delay. The research concludes that integrating computer vision significantly enhances navigation capabilities, opening up new possibilities for line-following robots in various applications.

Aharari & Ueda (2019) presented a method to improve the performance of line-following robots by reducing light noise interference using a low pass filter (LPF) applied to the color sensors of the robot. The key contributions of the article include the integration of LPF for smoothing out sensor readings, real-time noise reduction, and experimental validation showing a significant reduction in deviation from the target line. The methodology involves system setup, calibration and normalization, real-time application of LPF, and experimental testing on a 1.5 m × 1.5 m course. The key results indicate improved stability and accuracy, as well as noise reduction. However, the study has limitations related to dependency on initial calibration and a limited scope of testing. The research concludes that integrating an LPF into the color sensor processing of a line-following robot significantly improves its performance. The study offers practical methods for enhancing the reliability of line-following robots and suggests future work exploring the application of this technique to other sensors and environments.

Conclusions

The literature analysis on line-following robots reveals a robust body of work that has extensively explored various aspects of this technology. Over the years, researchers have primarily focused on control algorithms, sensor integration, and noise reduction techniques. Key developments include applying advanced control strategies such as sliding mode control (Yildiz et al., 2020) and integrating computer vision for enhanced path tracking (Prakash et al., 2012). Additionally, efforts to improve sensor accuracy through methods like the low pass filter for color sensors (Aharari & Ueda, 2019) have significantly enhanced the reliability of these robots in diverse environments.

In the past 3 years, the research trends have shifted towards integrating artificial intelligence and machine learning into line-following robots, as seen in studies exploring deep-learning approaches for motion control (Farkh et al., 2021). There has also been a growing interest in optimizing the performance of these robots through multi-objective optimization techniques (Gawlicki & Jankowski, 2021), through simulation (Brigido, Maximo & Parente de Oliveira, 2024) and exploring novel applications in industrial settings, such as integrating robotic arms with line-following robots for material handling tasks (Gosim et al., 2012).

There are potential future research opportunities, as outlined below: Scalability studies: – Conduct comprehensive studies on the scalability of line-following technologies, particularly for large-scale industrial environments.

– Explore ways to integrate AI-driven control systems effectively, focusing on optimizing processing efficiency and real-time decision-making.

– Comment: Addressing scalability is crucial for applying these technologies to industrial and transportation settings.

Enhancing environmental robustness: – Investigate the impact of environmental variations, such as extreme lighting conditions and diverse surface textures, on the performance of line-following robots.

– Develop and test techniques to improve robustness against these variations.

– Comment: Improving environmental robustness will ensure consistent performance across diverse real-world scenarios.

Dynamic and unpredictable environments: – Focus on the application of line-following robots in dynamic and unpredictable environments where real-time adaptability is essential.

– Develop sophisticated algorithms capable of handling complex and changing pathways effectively.

– Integrate additional sensor modalities, such as LIDAR and advanced camera systems, to improve adaptability and perception.

– Comment: Advancing these capabilities could significantly enhance the utility of robots in dynamic industrial and transportation contexts.

Broader applications: – Combine advancements in scalability and robustness to expand the potential applications of line-following robots in industrial, transportation, and other real-world scenarios.

– Comment: Addressing these challenges holistically will pave the way for next-generation robotic systems.

In conclusion, while line-following robotics has seen substantial growth, particularly in applying advanced control systems and AI integration, several areas remain ripe for exploration. Future research should focus on scaling these technologies for broader industrial applications, improving their adaptability in dynamic environments, and continuing to innovate in integrating intelligent systems to push the boundaries of what these robots can achieve.

Supplemental Information

Supplemental Information 1 The code used for analysis.

Supplemental Information 2 PRISMA checklist.

Additional Information and Declarations

Competing Interests

The authors declare that they have no competing interests.

Author Contributions

Williamson Johnny Hatzinakis Brigido conceived and designed the experiments, performed the experiments, analyzed the data, performed the computation work, prepared figures and/or tables, and approved the final draft.

Jose M. Parente de Oliveira performed the experiments, analyzed the data, authored or reviewed drafts of the article, and approved the final draft.

Data Availability

The following information was supplied regarding data availability:

The code used for the complete analysis is available in the Supplemental File and at GitHub: https://github.com/JohnnyEngineer/LineFollowerRobotLiteratureReview.

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
