# Peer review of "The line follower robot: a meta-analytic approach"

_PeerJ Computer Science, doi:10.7717/peerj-cs.2744_

## Round 0.1 · original submission · Major Revisions

Dear authors,
You are advised to critically respond to all comments point by point when preparing an updated version of the manuscript and while preparing for the rebuttal letter. Please address all comments/suggestions provided by reviewers, considering that these should be added to the new version of the manuscript.

Kind regards,
PCoelho

Reviewer 1 ·

Basic reporting

- The abstract of this work is well-written and informative. It provides a strong foundation for the full paper, which should delve deeper into the specific findings, methodologies, and implications of the research. By addressing the suggested improvements, the authors can further enhance the impact of their work.
- The work is generally well-written and concise. It clearly states the contribution of the article to the field of line-following robotics.

Experimental design

However, there are a few minor points that could be improved:
- The R software did not mention in abstract and I think need to explain it also in brief to know why you use it in this work?
- The field of line-following robotics has evolved significantly in recent years, with a strong emphasis on AI and Machine Learning Integration and Real-World Applications, so I need authors to focus on these applications in context.

Validity of the findings

By addressing following research gaps and exploring new avenues, Then this work will be a model to be emulated:
* Enhance Scalability: Scale AI-driven control systems for large-scale applications.
* Improve Environmental Robustness: Develop techniques to handle variations in lighting and surface conditions.

Cite this review as
Anonymous Reviewer (2025) Peer Review #1 of "The line follower robot: a meta-analytic approach (v0.1)". PeerJ Computer Science

Reviewer 2 ·

Basic reporting

This is an interesting paper reporting the literature approach TEMAC to the key contributions, trends, and gaps in the Line-follower robots.
If data in Section 0.2 could be presented in graph or chart, that would be easier to read and compare.

Experimental design

no commnet

Validity of the findings

The validation is basically fine. Since the authors has mentioned that not all articles could be found due to limited funded, otherwise, a wider literature would make the results more convincible.

Additional comments

This is an up-to-date study on how to make effective academic literature using the latest tools. If the research gaps could be described in a dot-list form and with added comments, that would be much helpful for researchers in this field.

Cite this review as
Anonymous Reviewer (2025) Peer Review #2 of "The line follower robot: a meta-analytic approach (v0.1)". PeerJ Computer Science

Reviewer 3 ·

Basic reporting

I appreciate the thorough review of the field of line-following robots. However, it is hard to claim that line-following robots are a valuable research subject. I am not convinced by the authors that this is a field that worth conducting a survey on. It is more of an engineering task than a research direction.

Experimental design

While I appreciate the immense effort and analysis, I do not see the survey being organized in any reasonable fashion. It feels like a pile of summaries of related papers.

Validity of the findings

Again, I do not find any of the findings/summaries contribute to the research community.

Additional comments

Please improve your figures. Most figures lack proper captions. Also, please do not use word clouds. They are the least scientific or efficient way to convey your idea. In additions, some fonts are too small to read.

Cite this review as
Anonymous Reviewer (2025) Peer Review #3 of "The line follower robot: a meta-analytic approach (v0.1)". PeerJ Computer Science

Reviewer 4 ·

Basic reporting

The English language is clear and unambiguous.

References and background context are provided sufficiently.

The review addresses the lack of literature of comprehensive analyses in the field of line follower robot, which is a important topic in the area of autonomous robots. Therefore it of interest within the scope of the journal.

According to the authors, only one systematic review has been identified for the topic and that review is yet limited to warehouse settings.

The introduction adequately makes the motivation clear.

Experimental design

The authors follow the principles of the Theory of the Consolidated Meta68 analytic Approach (aka TEMAC) methodology, makes the analyses systematic.

Sources are adequately cited.

The organization of the review follows the TEMAC method and is clear to follow.

Validity of the findings

The review clearly describes the key contributions, trends, and gaps in the field of line follower robots.

Conclusions are well stated and meet the goals in the introduction.

Additional comments

The authors should consider enlarge the text in most figures (e.g. 2-4, 6-10).

Cite this review as
Anonymous Reviewer (2025) Peer Review #4 of "The line follower robot: a meta-analytic approach (v0.1)". PeerJ Computer Science

---

## Round 0.2 · accepted · Accept

Dear authors, we are pleased to verify that you meet the reviewer's valuable feedback to improve your research.

Thank you for considering PeerJ Computer Science and submitting your work.

Kind regards
PCoelho

Reviewer 1 ·

Basic reporting

No further note

Experimental design

no further note

Validity of the findings

no further note

Cite this review as
Anonymous Reviewer (2025) Peer Review #1 of "The line follower robot: a meta-analytic approach (v0.2)". PeerJ Computer Science

Reviewer 4 ·

Basic reporting

The authors have addressed my concerns.

Experimental design

no new commnets

Validity of the findings

no new commnets

Additional comments

no new commnets

Cite this review as
Anonymous Reviewer (2025) Peer Review #4 of "The line follower robot: a meta-analytic approach (v0.2)". PeerJ Computer Science